# Emissions and fate of organophosphate esters in outdoor urban environments

Timothy F. M. Rodgers [1], Amanda Giang [1,2] ✉, Miriam L. Diamond[3,4], Emma Gillies [1] & Amandeep Saini [5] ✉

Cities are drivers of the global economy, containing products and industries that emit many chemicals. Here, we use the Multimedia Urban Model (MUM) to estimate atmospheric emissions and fate of organophosphate esters (OPEs) from 19 global mega or major cities, finding that they collectively emitted ~81,000 kg yr$^{-1}$ of $\Sigma_{10}$OPEs in 2018. Typically, polar "mobile" compounds tend to partition to and be advected by water, while non-polar "bioaccumulative" chemicals do not. Depending on the built environment and climate of the city considered, the same compound behaves like either a mobile or a bioaccumulative chemical. Cities with large impervious surface areas, such as Kolkata, mobilize even bioaccumulative contaminants to aquatic ecosystems. By contrast, cities with large areas of vegetation fix and transform contaminants, reducing loadings to aquatic ecosystems. Our results therefore suggest that urban design choices could support policies aimed at reducing chemical releases to the broader environment without increasing exposure for urban residents.

Cities are hotspots of human dynamism, culture, and industry, containing more than half of the world's population and generating over 80% of global GDP[1]. This concentration of people, products, and activities means that cities act as emissions sources for many chemicals, exposing urban residents, surrounding communities, and ecosystems to high levels of many chemical pollutants[2]. Understanding the dynamics of chemical emissions and fate in cities is therefore essential for reducing chemicals exposure, and helping us build "Sustainable Cities and Communities" (United Nations Sustainable Development Goal 11).

The control of persistent organic pollutants (POPs), for example through the Stockholm Convention[3], has focused on chemicals with persistent, bioaccumulative, and toxic (PBT) properties[4]. More recent work has recognized that although persistent, mobile, and toxic (PMT) organic chemicals do not bioaccumulate, they also pose a hazard, as they are not easily removed from water through traditional sorptive treatment processes and are therefore able to

contaminate surface, ground, and drinking water resources[5,6]. By definition, a less bioaccumulative substance will be more hydrophilic and mobile in water. Regulations aimed at controlling the use and release of PBT substances are therefore much less effective for PMT substances[5]. This can be one cause of "regrettable substitution," whereby chemical manufacturers respond to regulations around PBT substances by using chemicals that are less bioaccumulative, yet have PMT characteristics. One example of this phenomenon was the replacement of the flame retardant polybrominated diphenyl ethers (PBDEs) after their listing by the Stockholm Convention in 2009 and 2017[7]. Organophosphate esters (OPEs) were used as drop-in replacements for PBDEs in many commercial products, including the more soluble chlorinated OPEs, some of which are PMT substances[5,8-10]. OPEs have been found to undergo long-range transport, to be persistent in the environment, and to have serious health impacts on exposed populations, leading them to be called regrettable substitutes for PBDEs[11].

[1]Institute for Resources, Environment and Sustainability, University of British Columbia, Vancouver, BC V6T 1Z4, Canada. [2]Department of Mechanical Engineering, University of British Columbia, Vancouver, BC V6T1Z4, Canada. [3]Department of Earth Sciences, University of Toronto, Toronto, ON M5S 3B1, Canada. [4]School of the Environment, University of Toronto, Toronto, ON M5S 3B1, Canada. [5]Air Quality Processes Research Section, Environment and Climate Change Canada, Toronto, ON M3H5T4, Canada. ✉e-mail: amanda.giang@ubc.ca; amandeep.saini@ec.gc.ca

OPEs are ubiquitous contaminants found in cities across the world at high levels in urban air[12–14] and water[15,16], with large (1–2 orders of magnitude) variations in air concentrations observed between cities[12]. These large concentration differences arise from differences in both emissions and in the fate of the compounds within cities. Chemical emissions in cities come from a wide variety of sources. Point-source emissions originate from industrial and manufacturing processes, while diffuse emissions originate from OPEs used in products. Depending on the chemical and the location, either point-source or diffuse emissions can dominate[17,18]. This wider variety of sources makes estimating OPE emissions difficult, with uncertainties that span orders of magnitude[18–20]. In places with large manufacturing bases such as Beijing, China, a combination of emissions from OPE production and manufacturing may be responsible for the majority of emissions[21]. In other areas where manufacturing plays a smaller role, such as Toronto, Canada, diffuse sources may dominate[8].

Urban environments tend to increase chemical mobility through the water. Urbanization is typified by large areas of impervious surfaces, which reduce the ability of natural sorptive processes (such as infiltration through a riverbank) that would otherwise capture contaminants[22]. The large area of impervious surfaces in urban environments accumulates an organic surface film[23] that further enhances the transport of semi-volatile organic compounds (SVOCs) from the atmosphere to surface compartments[2,24]. The films capture gas- and particle-phase SVOCs that are transferred by rainwater to soils and into urban waterways[25]. Thus, cities are important starting points for the global long-range transport of chemicals through air and water[8,26]. A changing climate is also affecting how chemicals move through the environment[27,28], by promoting more release to warmer air, more water-borne transport in locations experiencing greater precipitation, and more atmospheric transport in locations experiencing drought.

Despite the importance of cities as sources of many chemicals, differences in chemical fate between urban environments have not been well-studied. Here, we address this gap by combining a unique dataset from the Global Atmospheric Passive Sampling (GAPS)-Megacities network[12] with the Multimedia Urban Model (MUM)[8,24] to investigate the emissions and fate of OPEs in 19 mega or major cities around the world. The goals of this study were to: (1) Estimate the emissions of OPEs in the 19 GAPS-Megacities locations, (2) Investigate the sources of those emissions, (3) Investigate how built-environment, physico-chemical properties, and climatic factors influence the fate of chemicals in different urban environments, and (4) Provide recommendations for policy or engineering solutions that could reduce chemicals emissions from cities.

## Results

### Air emissions estimation and model evaluation

We estimated aggregate air emissions by back-calculating the emissions required to maintain the reported air concentrations from the 19 cities under the GAPS-Megacities study[12], using an instantiation of MUM parameterized for each city across the ~3-month sampler deployment period (Fig. 1).

Overall, we estimated that the 19 cities in our study emitted 81,000 kg yr$^{-1}$ $\sum_{10}$OPEs (Fig. 2) to the air in 2018. Estimated emissions varied by nearly 40-fold between cities. London had the largest emissions at ~39,000 kg yr$^{-1}$, followed by Bogotá at ~13,000 kg yr$^{-1}$, while Sydney, Kolkata, and Istanbul all had <100 kg yr$^{-1}$ of $\sum_{10}$OPE emissions.

On a compound-specific basis (Supplementary Table 1 contains the names and identifiers for all compounds modeled in this study), emissions of tris (1-chloro-2-propyl) phosphate (TCIPP) were the largest, at ~53,000 kg yr$^{-1}$, followed by tris(2-chloroethyl)phosphate (TCEP), at ~15,000. Together, these two compounds accounted for ~85% of all estimated $\sum_{10}$OPE emissions. In every city, either TCIPP or TCEP had the largest emissions, and combined they comprised 48–91% of emissions in each city.

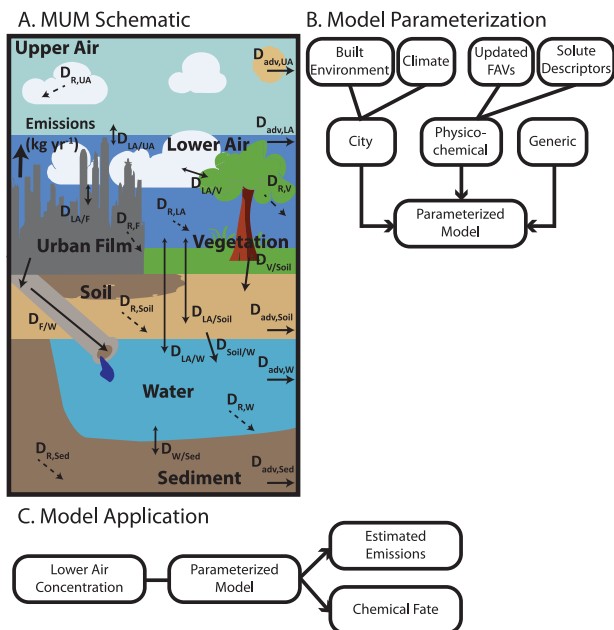

**Fig. 1 | Modeling framework. A** Schematic diagram of the Multimedia Urban Model (MUM) showing the seven compartments (upper air (UA), lower air (LA), urban film (F), vegetation (V), soil (Soil), water (W), and sediment (Sed)); inter-compartmental transport processes (solid arrows, $D$ values (mass/time) with compartment subscripts); emissions to air; transformation processes (dashed arrow, $D_R$); and advective transport out of the system ($D_{adv}$). Bi-directional processes are shown with double-headed arrows, with the larger arrow showing the typical direction of net mass transport. **B** Flowchart showing the model parameterization, where FAVs refer to final adjusted values. **C** Flowchart showing the model application for an individual city. The tree, grass tufts, clouds, and city skyline were generated with the assistance of DALL·E 2[89].

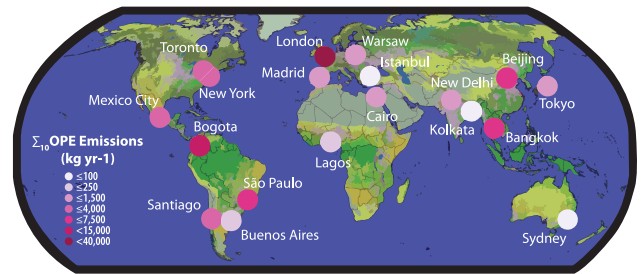

**Fig. 2 | Map showing 2018 $\sum_{10}$ organophosphate ester (OPE) air emissions from the 19 Gaps-Megacities locations.** Emissions were calculated using administrative boundaries. The base map shows global land cover from Copernicus Global Land Service[60] overlaying country borders from the Global Administrative Data Map[82].

Based on the comparisons presented here and the full MUM uncertainty analysis of ref. [8], our emissions estimates have approximately an order of magnitude uncertainty in either direction for each city. The 2018 $\sum_{10}$OPEs predicted emissions were similar to previous estimates, which were available for the city of Toronto and for Beijing at a provincial level. In Toronto, the 2018 emissions were ~45% lower than the emissions predicted by ref. [8]. for 2010 using the same model, with most of the difference caused by the lower air concentrations used here. In Beijing, our estimates for the municipal area were ~50% lower on an area-normalized basis than the provincial estimates of ref. [21]. for 2018. Their estimated air concentrations were close to those input here to back-calculate the emissions, meaning that the difference in emissions intensity was likely caused by different estimations of chemical fate within the modeled domain. Further, our predicted

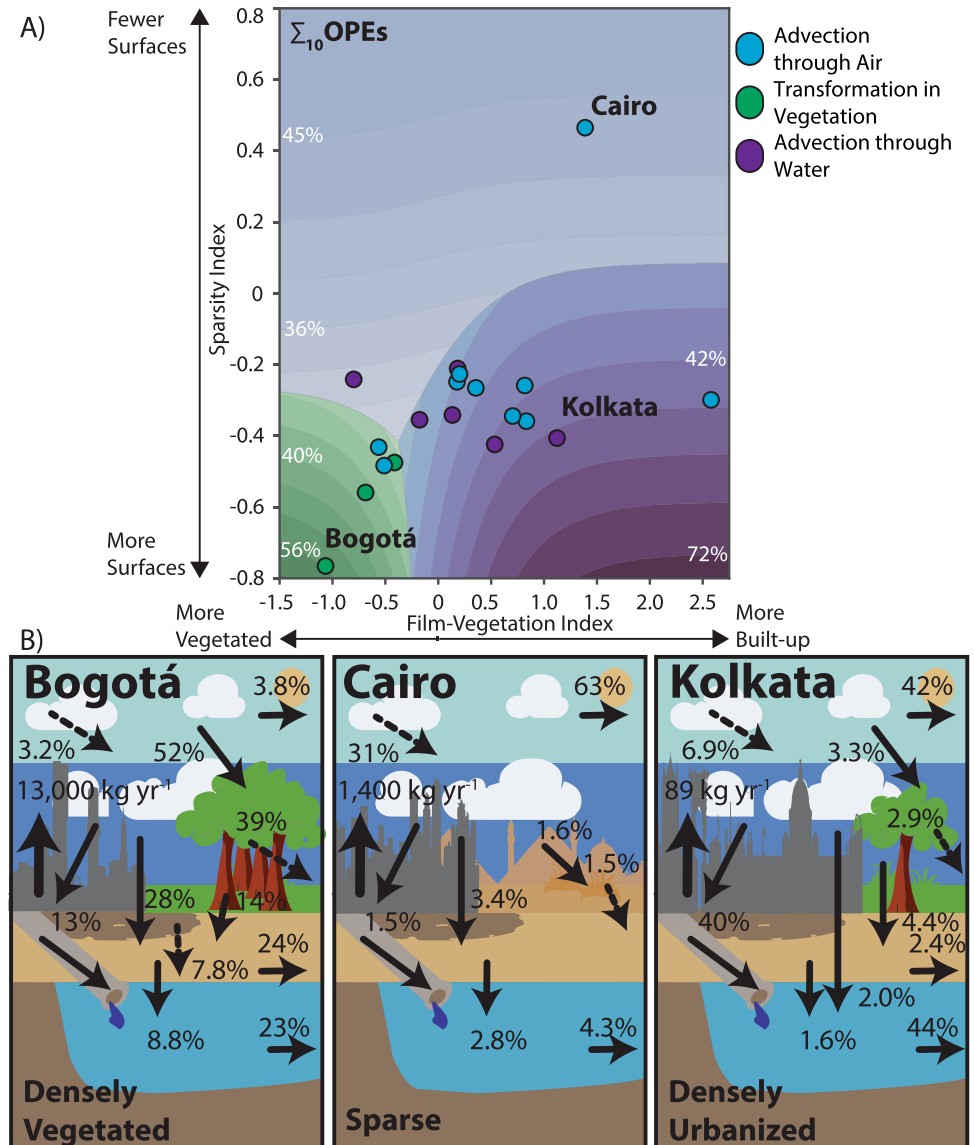

**Fig. 3 | Influence of the built-environment on ∑10 organophosphate ester (OPE) fate. A** City-space figure showing the dominant chemical fate process for the $\sum_{10}$OPEs with their built-environments described by the surfaces vs film-vegetation indices. Contour colors show the dominant fate process, with the intensity showing the proportion of total emissions undergoing that process (as labeled). Points show where the 19 GAPS-Megacities locations fit on these axes; the point color represents the 2018 dominant fate process in each city. **B** $\sum_{10}$OPE fate diagrams for Cairo,

Bogotá, and Kolkata, respectively, for 2018. Dashed lines represent transformation processes and solid lines transport processes. Emissions (kg yr$^{-1}$) are shown entering the lower-air compartment and fate process values are given as the % of total emissions. Values shown on each figure may not sum to 100 as only larger processes are shown. The trees, grass tufts, clouds, and city skylines were generated with the assistance of DALL·E 2[89].

concentrations in media other than air were generally within a factor of 100 of published measurements in those same media (Supplementary Fig. 1 and Supplementary Results S1), comparable to the accuracy of predictions of remote air concentrations made using the BETR-Global model for PBDEs[19] and to the agreement between predicted and measured soil concentration across China for OPEs[21].

### Identifying drivers of OPE emissions

One of our central goals was to assess whether we could identify the sources or sectors that contribute to OPE emissions, and if we could use our results to develop proxies for OPE emissions in the absence of measured inventories. We, therefore, correlated the log$_{10}$-transformed emissions flux (log$_{10}$ kg m$^{-2}$ yr$^{-1}$) with several proxies for emission sources (Supplementary Fig. 4 and Supplementary Table 3). For instance, we used gross domestic product (GDP, 2015 \$ at purchasing power parity)[29] and population[30] to estimate broad-based emissions

from in-use products, and we used sector-specific estimates of anthropogenic greenhouse gas emissions[31] to estimate contributions from various industrial sectors.

Our results suggested that at a global scale, most OPE emissions originate from numerous complex, diffuse sources, rather than from specific manufacturing or production processes. The strongest single correlation was with ∑GDP in the modeled area, which explained 36% of the variation (measured by $r^2$) for the log$_{10}$ $\sum_{10}$OPEs, driven by correlations ($r^2$ of 0.31–0.19, $p < 0.05$) for, in descending order, tris(3-methylphenyl) phosphate (TmCP), tributyl phosphate (TnBP), TCEP, TCIPP and tris(1,3-dichloroisopropyl) phosphate (TDCIPP) (Supplementary Fig. 4). Most individual correlations between emissions and sector-specific proxies were weak ($p > 0.05$, Supplementary Table 3). For TCIPP, which is used extensively in building insulation[32,33], diffuse emissions from building materials appeared to be a major source, with log$_{10}$ emissions moderately correlated with the percentage of

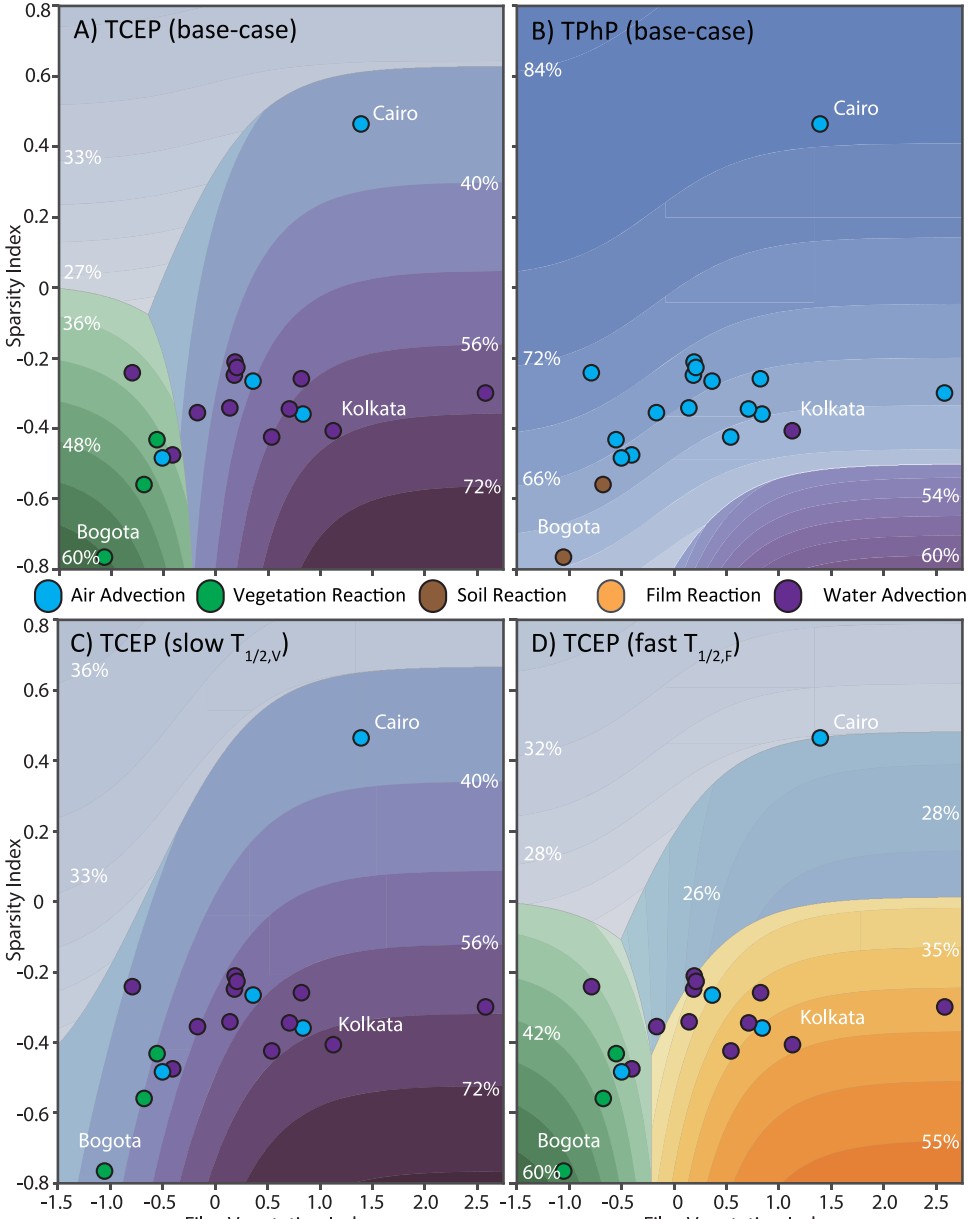

**Fig. 4 | City-space diagrams showing the influence of physicochemical properties on organophosphate ester (OPE) fate.** Dominant chemical fate processes for **A** tris(2-chloroethyl)phosphate (TCEP) and **B** triphenyl phosphate (TPhP) using the average city parameterization, **C** TCEP with the vegetation reaction half-life ($T_{1/2,V}$) slowed by a factor of 10 and **D** TCEP with the film reaction half-life ($T_{1/2,F}$) quickened by a factor of 100. Points represent the 19 GAPS-Megacities locations; the color of each point represents the dominant fate process for that chemical in each city using its 2018 parameterization. Contour colors represent the dominant fate process in each region, with the intensity of the proportion of total emissions undergoing that process (as labeled in each region). Note that reaction in soil was the dominant process for TPhP in two cities but does not show on the contour plots using the "average" parameterization.

greenhouse gas emissions (% of $CO_2$ equivalent $kg\,m^{-2}\,s^{-1}$) from the "energy for buildings" and the "solvents and other products use" (a broad-based measure of in-use products) categories ($r^2 = 0.28$ and 0.35, respectively). Supplementary Results S2 contains additional information on the correlations, including Supplementary Table 3 with all regression statistics.

**Fate of organophosphate esters in outdoor urban environments**
Our results showed that contaminant fate processes had a large impact on environmental concentrations, and therefore both the magnitude and the pathways for human and ecosystem exposures. Our sensitivity analysis (Supplementary Fig. 3 and Supplementary Table 3) indicated that there were three groups of parameters, which collectively controlled contaminant fate in outdoor urban environments: those

representing the built environment, physicochemical properties, and climate. We investigated the relationships among these groups of parameters by running the model for several scenarios across a city-space which represented different cities by their "sparsity index" and "film-vegetation index". As described in the Methods, we built these two indices to represent three critical built-environment drivers of chemicals fate: the city's footprint ($A_{city}$, $m^2$), the area factor of the vegetation compartment ($AF_{veg}$, $A_V/A_{city}$, $m^2\,m^{-2}$), and the area factor of the urban film compartment ($AF_{film}$, $A_F/A_{city}$, $m^2\,m^{-2}$). We defined this "sparsity index" ($m^2\,m^{-2}$) with Eq. (3):

$$\text{Sparsity Index} = \log\left(\frac{A_{\text{city footprint}}}{A_{\text{film}} + A_{\text{vegetation}}}\right) \quad (1)$$

where $A_j$ represents the area of compartment j in m². The second index explored the nature of those surfaces. We defined this "film-vegetation index" with Eq. (4):

$$Film - Vegetation\ Index = \log\left(\frac{A_{film}}{A_{vegetation}}\right) \quad (2)$$

First, we looked at the influence of the built-environment alone by running our model using a synthetic "average" city, with the model parameters (outside of those in the sparsity and film-vegetation indices) and the input concentrations for each of the OPEs set to their mean values across the 19 cities (Fig. 3A). Next, we looked at the influence of physicochemical properties by contrasting the fate of a polar PMT-like compound, TCEP, with a non-polar PBT-like compound, triphenyl phosphate (TPhP), across the same average city-space and for scenarios exploring transformation half-lives. Finally, we probed the influence of climate by running the model across the city-space with the average city climate replaced by composite "low-deposition" and "high-deposition" climates for the PMT-like and the PBT-like compound.

## Influence of the built environment

Across our city-space diagram (Fig. 3A), cities with a high sparsity index have fewer depositional surfaces, while cities with a low sparsity index have more surfaces. The film-vegetation index describes the nature of those depositional surfaces. For cities with a film-vegetation index >0, the area of the urban film is greater than the area of vegetation, and vice-versa.

$\sum_{10}$OPE fate varied substantially between three city archetypes: "Sparse," "Densely vegetated," and "Densely urbanized," represented by Cairo, Bogotá, and Kolkata, respectively (Fig. 3B, Supplementary Figure 5 shows the $\sum_{10}$OPE fate diagrams for all 19 cities, and SI Figs. S6–S15 show the fate diagrams for each compound across all 19 cities). In sparse cities with fewer depositional surfaces (Fig. 3A, blue-shaded contours), such as Cairo (Fig. 3B), $\sum_{10}$OPE fate was dominated by air advection from the city to its surrounding region. In our dataset, Cairo was the only city where the area of film and vegetation surfaces was lower than the area of the city's footprint, due to the large area of bare-ground, and this led to ~94% of the $\sum_{10}$OPEs emissions remaining in the air compartment and either undergoing primary transformation or being blown down-wind.

In cities with many surfaces (low sparsity index) like Bogotá (vegetation) and Kolkata (urban film), deposition played a much more significant role, with up to 93% of emitted chemicals deposited to surfaces within Bogotá's city limits. The fate of the compounds deposited was then determined by the nature of the depositional surfaces. In "densely vegetated" cities (Fig. 3A, green-shaded contours), represented by Bogotá (Fig. 3B), deposition to and subsequent transformation in the vegetation compartment dominated chemical fate. Plants are able to take-up and metabolize some OPEs[34–36], so the vegetation compartment here acted to fix the compounds in place and transform them. In densely-vegetated Bogotá, 39% of overall OPE mass was transformed in the vegetation compartment, while 14% was predicted to be washed off into the soil. Further, we predicted that 24% of overall atmospheric OPE emissions would either be buried in the soil or infiltrate into groundwater, highlighting an important risk with PMT chemicals.

In densely urbanized cities (Fig. 3A, purple-shaded contours) with very high impervious surface coverage, like Kolkata (Fig. 3B), OPE fate was dominated by deposition to film followed by wash-off through stormwater and subsequent advection from the city to the surrounding aquatic ecosystem. Thus, water advection accounted for the fate of ~44% of emissions, with 42% lost via wind advection. This is less than the >56% water advection that would be predicted using the characteristics of the average city, due in part to the physicochemical

properties of the OPEs released in Kolkata and in part to its climate, as will be explained below.

## Influence of physicochemical properties

We compared the fates of individual OPEs to assess the influence of physicochemical properties, using TCIPP and TPhP as chemicals with representative mobile and bioaccumulative behavior, respectively (Fig. 4A, B and S1 Figs. S2–S11 show the base-case fate of each compound). Low $K_{OW}$, soluble PMT-like compounds such as TCEP required fewer surfaces for deposition to dominate due to their higher solubilities leading to more atmospheric wash-out. Conversely, for the higher $K_{OW}$, lower solubility PBT-like compounds, represented here by TPhP, less efficient scavenging from precipitation meant that more surfaces were required for atmospheric deposition to take place. Thus, air advection dominated across almost all cities, with water advection being considerably less important than for the PMT-like compounds, as expected.

For OPEs and other compounds with shorter transformation half-lives in vegetation (i.e., that were susceptible to phytotransformation), plants acted as fixing and transforming surfaces, reducing the concentration of OPE parent compounds that either remained in the air compartment or were exported to aquatic ecosystems. Although direct atmospheric transformation products of OPEs can be more persistent and toxic than the parent compounds[37], plants have been shown to rapidly transform the predominantly triester OPE parent compounds primarily through direct dealkylation to diester products, or through hydroxylation to hydroxylated OPEs[38]. Subsequent transformation of the diester products has been observed for the non-chlorinated OPEs[3]. This continued metabolism suggests that plant transformation may be able to reduce the overall persistence of non-chlorinated OPEs and their transformation products, thereby lowering the overall hazard posed by OPEs deposited to plants. By contrast, for the chlorinated OPEs, the lack of continued metabolism indicates that the transformation products may continue to be problematic. For two cities (Bogotá and Mexico City), the reaction in the soil dominated the overall fate of TPhP, following chemical deposition to vegetation and subsequent wash-off to the soil, as TPhP is less susceptible to transformation in vegetation than in soil.

The amount of transformation in the vegetation compartment was sensitive to the modeled transformation half-life, meaning that compounds that are only slightly less susceptible to phytotransformation are unlikely to be transformed by plants, and for those compounds, plants will be less effective at fixing and transforming contaminants rather than mobilizing them. Slowing the vegetation transformation half-life ($T_{1/2,V}$) by a factor of 10 (to represent hypothetical compounds more resistant to or slower at transformation) removed plant transformation as a dominant process (Fig. 4C shows the city-space diagram for TCEP under these conditions), with most of the mass deposited to plant surfaces either re-volatilizing to air and leaving the city through air advection, or washing through to soil to the water compartment and then advecting downstream; for some compounds, this also increased transfer to groundwater.

By contrast, the urban film mobilized OPEs by enhancing their transfer to the water compartment and increasing loadings to aquatic ecosystems. The urban film consists of a mixture of organic matter, soot, and deposited atmospheric particles that accumulate over time, thus giving it complex chemical characteristics[25,39,40]. Surface-mediated chemical reactions on urban films or particles can be important for some chemicals[40,41], but OPEs are generally believed to have up to order-of-magnitude lower reaction rates when particle-bound due to the ability of particles or atmospheric water to shield OPEs from hydrolysis[37,42,43].

Fate in the film compartment was less sensitive to the transformation half-life ($T_{1/2,F}$) than fate in the vegetation compartment, as a similar 10x decrease in $T_{1/2,F}$ did not change the dominant fate

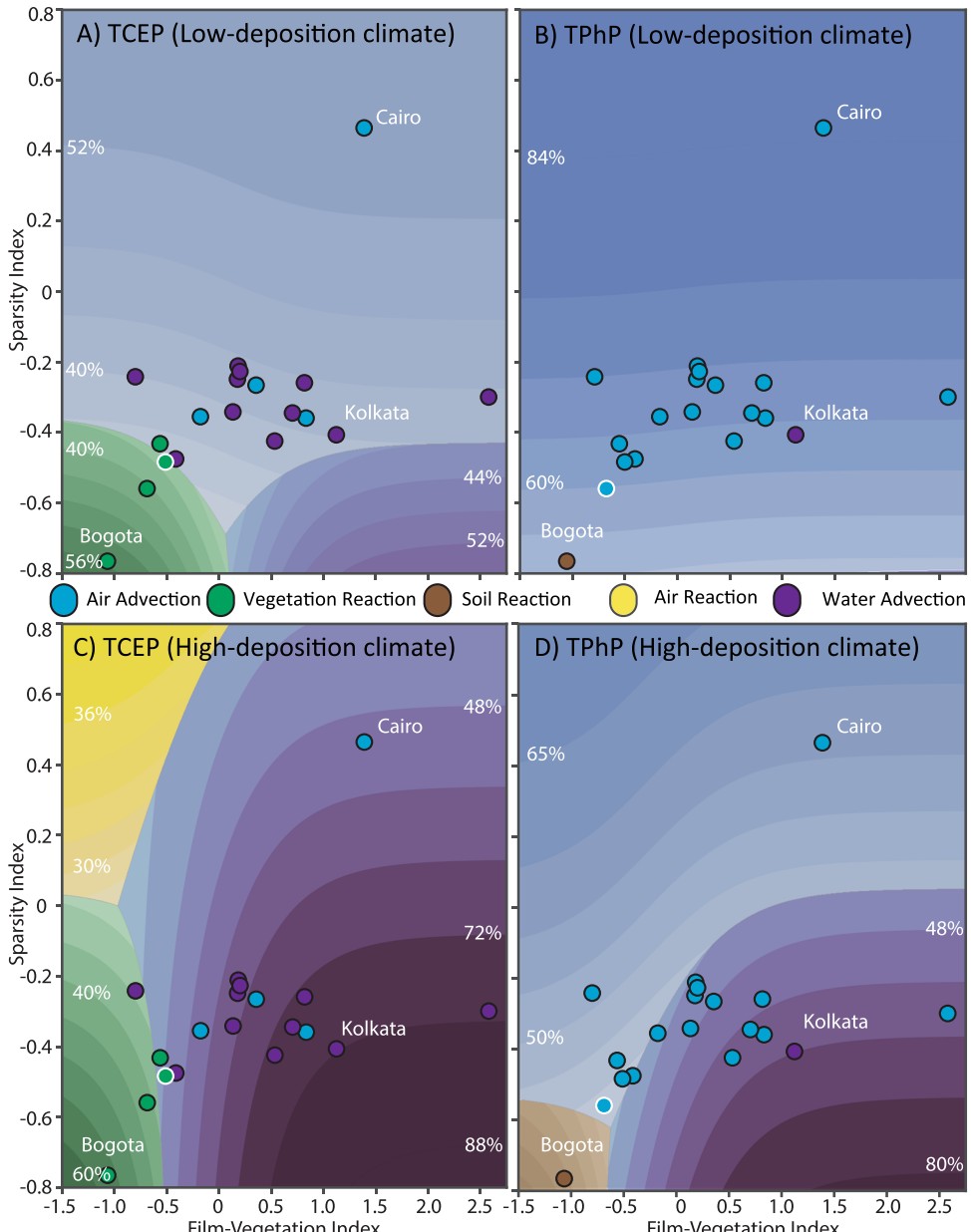

**Fig. 5 | City-space diagrams showing the influence of climate on OPE fate.** Fate of **A** TCEP and **B** TPhP using the low-deposition city parameterization. Fate of **C** TCEP and **D** TPhP using the high-deposition city parameterization, as described in the main text. Points represent the 19 GAPS-Megacities; the color of each point represents the dominant fate process for that chemical in each city using its SSP3- 7.0 2100 parameterization, with white outlines highlighting those that changed from the 2018 baseline. Contour colors represent the dominant fate process in each region, with the intensity of the proportion of total emissions undergoing that process (as labeled in each region).

processes across the city-space diagrams. However, increasing $T_{1/2,F}$ by 100x (likely a maximum rate, although the reaction rate in the urban film is poorly constrained) led to a transformation in the film compartment dominating (Fig. 4D). Thus, the film compartment is more likely to transfer chemicals to water rather than fix and transform them.

### Influence of climate

Inter-city climatic variability was mainly responsible for the differences seen between the "average" cities (contour lines) and the fate in individual cities (filled-in circles) in Figs. 3, 4. Across the city-space, a low-deposition climate was warmer, drier, and windier, with a higher planetary boundary layer height, and cities with this climate tended to be dominated by air advection (Fig. 5A, B). A high-deposition climate was cooler, wetter, and calmer, with a lower ceiling, and cities with this

climate tended to be dominated by vegetation reaction and water advection (Fig. 5C, D). The low-deposition climate was parameterized using the 5th percentile lowest precipitation rate observed across the 19 megacities and the 95th percentile highest windspeed, temperature, and planetary boundary layer height, while the high-deposition climate was parameterized inversely.

The fate of even the same chemical in the same built environment was substantially different between the low-deposition and the high-deposition climates (Fig. 5). This meant that, depending on the climate, traditionally waterborne PMT-like chemicals such as TCEP could be advected via air rather than water, and traditionally sorptive PBT-like chemicals such as TPhP could become water-borne contaminants. Under the warmer, windier, and drier low-deposition climate, advection via air would dominate across almost all of the cities (Fig. 5A, B) for both the soluble PMT-like compound TCEP and the sorptive PBT-like

compound TPhP. By contrast, in the cooler, wetter, and calmer high-deposition climate, water advection and vegetation reaction were predicted to dominate across all the cities for TCEP, and water advection dominated for most (~12/19) of the megacities for TPhP (Fig. 5C, D).

The "SSP3-7.0" projected 2100 climatic differences between a more aggressive climate-change mitigation pathway with low emissions (SSP1-2.6) and a less aggressive mitigation pathway with higher emissions (SSP3-7.0) did not substantially change projected chemical fate across the conceptual city-space using the average city. Localized changes did, however, change the dominant fate processes for individual cities (colored circles with a white border, Fig. 5).

## Discussion

First, our results confirm that cities are important sources of OPE emissions. Further, we found that emissions of OPEs likely dwarfed emissions of the PBDEs that they replaced. The population of the 19 megacities presented here represents ~13% of the global population in cities with a population larger than 500,000[1]. We estimated $\sum_{10}$OPE emissions of between 3.8–7000 mg capita$^{-1}$. Extrapolating these values to the global urban population implies that cities with a population of >500,000 could emit 0.88–140 (mean of 16) kt yr$^{-1}$ of $\sum_{10}$OPEs. This compares with a total of 9.3–25 (mean of 16) kt of PBDEs estimated to be emitted since production began in the 1970s[19].

Second, emissions across cities appeared to be driven more by diffuse, economy-wide processes than individual manufacturing sectors, as represented by proxies. We identified that a city's total GDP was the overall best proxy for OPE emissions. This indicates that OPE emissions come from a profusion of complex, distributed sources, making engineered solutions on manufacturing facilities unlikely to have much impact on overall OPE emissions.

Third, our results showed that both the built environment and climate strongly influenced chemical fate. Strikingly, the difference in the fate of a single chemical between cities with different climate and built environment factors was of a similar magnitude to the difference between a PMT-like and a PBT-like chemical in the same environment. Chemicals management tools and regulatory approaches generally screen chemicals for hazard traits (such as bioaccumulation or mobility) using their physicochemical properties, and the tools used to support chemicals management regulations often consider a single evaluative environment, such as is the case for the OECD Tool[44] or the evaluative multimedia environment in the Estimations Program Interface (EPI) Suite of software tools[45]. Our results indicate that in order to take a precautionary approach, regulatory support tools should consider that in different plausible emissions environments, the same chemical may appear to be mobile or bioaccumulative. To account for this influence of climate and the built environment on chemical fate, more weight could be placed on persistence and toxicity as hazard traits than on mobility and bioaccumulation.

Fourth, our results indicate that densely urbanized, sparsely vegetated cities in non-arid environments are extremely efficient at mobilizing chemicals to water through stormwater, and this means that more chemicals are likely to be found in stormwater than might be expected based on physicochemical properties alone. Recent work has highlighted the need for more green infrastructure to treat a wide variety of pollutants[22]. Our results suggest that diverting stormwater runoff from directly entering receiving bodies could significantly reduce aquatic loadings. Depending on the local context, this green infrastructure could range from engineered systems like bioretention cells to a simpler redirection of stormwater from rooves to, for example, gardens or other vegetated areas. Encouragingly, sorption-based green infrastructure technologies are effective for compounds with log $K_{OW}$ > ~3.8[46], meaning that for many of the more hydrophobic chemicals mobilized by cities (that would not be released to water in non-urban environments), green infrastructure should be an effective

way to decrease loadings to aquatic ecosystems. One additional note of optimism is that our results suggest that increasing the amount of green space in a city can increase a city's urban metabolism, directly removing chemical contaminants from the air and prevent them from being washed into the water, at least for those compounds that phytotransform into less toxic products.

Finally, the processes governing OPE emissions and fate in urban areas have significant implications for human and ecosystem exposure. Both emissions and urban design levers could therefore affect these exposures, though further research is needed on the impacts of different interventions. People are exposed to OPEs mainly via diet, dust ingestion, and dermal absorption (for toddlers); and via diet, indoor air inhalation, and dermal absorption (for adults); with drinking water a less studied but potentially significant pathway for the mobile chlorinated OPEs[47]. Designing our built-environments to favor certain processes over others will therefore involve complicated tradeoffs between exposures to different groups, and will require further investigation. For instance, as most food production occurs outside of cities, processes which act to retain OPEs in urban areas are likely to reduce human exposure via diet. However, if these processes simply mobilize OPEs to surface water, they will increase human exposure through drinking water, especially for the chlorinated OPEs, which are poorly removed by water treatment systems[46,48,49] and therefore may accumulate in water cycles[5]. Aquatic ecosystems are believed to be sensitive to certain OPEs[50], so moving OPEs from the atmosphere to water would also increase environmental damage. Further research to better understand these tradeoffs will allow us to design cities to better "metabolize" OPEs and other contaminants, preventing exposure for people and ecosystems within and outside of urban areas. Ultimately, our results suggest that supplementing policies that reduce sources of emissions with careful urban design to fix and transform (or metabolize) those chemicals we cannot eliminate provides the best pathway toward building healthier, more sustainable cities.

## Methods
### Model approach

The "Multimedia Urban Model"[24] (MUM) is a multimedia fugacity-modeling tool that accounts for urban contaminant dynamics in a steady-state, city-scale modeling domain (Fig. 1). It has been used to estimate levels of PAHs, PCBs, and PBDEs[51,52]. We used a version of the model that was parameterized for PMTs and used to estimate the emissions of OPEs from Toronto[8]. MUM follows the fugacity ($f$, Pa) multimedia modeling approach popularized by Mackay[53].

In this approach, the environment is broken into different compartments (e.g., air or water), and the concentration ($C$, mol m$^{-3}$) of a chemical in a compartment is defined as $C = f*Z$, where $Z$ (mol m$^{-3}$ Pa$^{-1}$) is the fugacity capacity. The fugacity capacity of air is defined as $Z_A = 1/RT$, where $R$ (J mol$^{-1}$ K$^{-1}$) is the ideal gas constant and $T$ (K) the temperature. The fugacity capacities of the other compartments are derived from the fugacity capacity of air using the partition coefficient ($K_{jk}$ for two phases $j$ and $k$) between air and that compartment. For instance, the fugacity capacity of pure water is $Z_W = Z_A/K_{AW}$, where $K_{AW}$ is the unitless air-water partition coefficient. Most environmental compartments are not made of pure substances, so the bulk fugacity capacity for a given compartment is calculated as the volumetrically-weighted sum of the fugacity capacities of the individual pure substances that are assumed to be at chemical equilibrium within that compartment. For instance, in MUM the fugacity capacity of bulk air ($Z_{A,B}$) consists of pure air ($Z_A$), pure water ($Z_W$), and aerosols ($Z_q$) so that $Z_{A,B} = Z_A \times VF_A + Z_W \times VF_W + Z_q * VF_q$, where the volume fraction for compartment j is denoted by $VF_j$. MUM consists of eight bulk compartments (Fig. 1): the lower air, upper air, water, soil, sediment, vegetation, and film. The full equations for each

compartment's fugacity capacity can be found in the code archived in our data repository[54], or in ref. [8].

MUM is a "level III" multimedia model, meaning that it is at temporal steady-state but that the different compartments are not at chemical equilibrium[53]. Chemical transport between compartments is modeled using "$D$ values" ($D_{jk}$, mol Pa$^{-1}$ h$^{-1}$) which define mass transport rates ($N_{jk}$, mol h$^{-1}$) between compartments $j$ and $k$ as $N_{jk} = f \times D_{jk}$. The overall $D$ values between compartments are found by the addition of different transport processes. At steady-state, the mass-balance equations for each compartment, consisting of $D$ values, fugacities, and sources to each compartment $j$ ($s_j$, mol hr$^{-1}$) can be combined into a single system of equations. For an air-water system, this would look like Eq. (3):

$$\mathbf{D}\vec{f} = \vec{s} = \begin{vmatrix} -D_{T,A} & D_{WA} \\ D_{AW} & -D_{T,W} \end{vmatrix} \cdot \begin{vmatrix} f_A \\ f_W \end{vmatrix} = \begin{vmatrix} s_A \\ s_W \end{vmatrix} \tag{3}$$

Where the total $D$ values leaving each compartment $j$ are denoted by $D_{T,j}$. Solving this system of equations, where the $D$ values and inputs are known, gives the fugacities in each compartment, allowing concentrations, masses, and mass transport to be calculated directly. We also used the model to back-calculate air emissions from measured concentrations. In this case, we first calculated the fugacity of the lower air compartment as $f_A = C_A/Z_{B,A}$, then moved all terms including $f_A$ to the right-hand side of the equation and the unknown emissions sources to air to the left-hand side, then solved the resulting matrix to provide the emissions to the lower air compartment and the fugacities of the other compartments. The full equations for each compartment's $D$ values can be found in the code archived in our data repository[54], or in ref. [8].

## Model parameterization

We parameterized the model for each of the 19 cities in the GAPS-Megacities network using a combination of remotely-sensed and locally available data. Datasets were processed using a combination of the numpy[55], xarray[56] and rioxarray[57] python packages, QGIS[58], and Google Earth Engine;[59] all of the code used in this analysis is available from the lead author's GitHub and our Data Repository. Our data repository[54] contains the values that were used as inputs to the model, the processed geospatial datasets that were used in this paper, or the code that can be used to obtain them. All continuous variables were clipped to the required city's model boundary using QGIS, taking either the mean value or the sum as appropriate.

We used the Copernicus Global Land Services 100 m Epoch 2018 land cover[60] as a basis to parameterize the dimensions of the model compartments, with the ground-area calculated as the total area of the model boundary multiplied by the percentage of pixels for each land use, using water for water, bare-ground plus the ground-area of vegetation for soil (representing soil underneath vegetation), all vegetation types as vegetation and built-up area for the urban film. We estimated the surface area of the vegetation using the leaf-area index (m² upwards-facing leaf-area per m² ground) multiplied by the ground-area. For the urban film, we used an analogous "impervious surface index" (ISI), defined as the ratio of the total surface area of impervious surfaces (e.g., building walls, roofs, roads, etc.), multiplied by the total built-up area. We were able to find detailed building footprints and heights for eight cities: Buenos Aires[61], Sydney[62], Toronto[63], Warsaw[64], Madrid[64], New York[65], São Paulo[66], and London[64]. For each of these cities, we calculated the impervious surface area for each building as the perimeter multiplied by the average building height plus the building footprint area. For datasets that were provided in raster format, we first converted the building footprints to a vector format with one vector object per building. The processed dataset with all eight cities is available in vector form from our data repository. We calculated the ISI for eight city administrative areas, five 5 km buffer areas, and two 15 km buffer areas where we could find detailed information on building heights and footprints. For the other city boundaries, we predicted the ISI using linear regression ($r^2 = 0.78$, $p < 0.01$) with the "built-up area density" (number of people per m² built-up area), a common metric of urban density[67] that we found provided the most stable predictions of ISI (Supplementary Fig. 4).

We obtained data on the leaf-area index, relative humidity (estimated from the dewpoint and surface temperature), windspeed (used to calculate the advective flow rate in the upper and lower air compartments), precipitation rate, and temperature from the Copernicus ERA5 Land ECMWF reanalysis dataset[68]. The height of the planetary boundary layer was used as the top of the upper air compartment, and was obtained from the Copernicus ERA5 ECMWF dataset[69]. We used a fixed height of 50 m for the height of the lower air as in ref. [8]. We obtained river flow rates from the GLOFAS ERA5 reanalysis (choosing the pixel or sum of pixels that appeared to accumulate each city's flow)[70], and river depths from ref. [71]. These were used to parameterize the flow rate and depth of the water compartment, with the area taken from the land cover dataset. In the air compartment, total suspended particle (TSP) concentrations were obtained from observed aerosol concentrations. Generally, TSP was not available so we used empirical relationships[72] to derive TSP from PM$_{10}$ or PM$_{2.5}$, using the largest size-fraction for which data were available. Some notable sources include the SPARTAN network[73] and the AirNow platform from US Embassies[74]. If no other data were available, we used a global PM$_{2.5}$ dataset by ref. [75]. All of the particulate matter data used is available in the Data Repository.

For chemical-specific parameters, where available, we used the recommended final adjusted values (FAVs) from ref. [76]. that incorporated measured and in silico estimations. We also calculated new FAVs for triethyl phosphate (TEP), tripropyl phosphate (TPrP), and tributyl phosphate (TnBP). Several of the OPE FAVs from Rodgers et al[76]. included $K_{OA}$ measurements made using an indirect technique that may show bias for more polar compounds[77–79]. As the FAV method adjusts the parameters of all of a compound's physicochemical properties based on their agreement, this bias in one property could propagate to all of the property values for a compound. An advantage of the Bayesian FAV method is that the prior distributions can be parameterized to incorporate our understanding of the uncertainty around the inputs in a transparent, reproducible manner. Since the indirect method is thought to produce $K_{OA}$ values that are biased low, we recalculated the FAVs for these compounds with a skew-normal distribution on the log $K_{OA}$ prior, increasing the probability that the model would adjust the $K_{OA}$ values upwards. We parameterized the polyparameter linear free energy relationships (ppLFERs) used by the model with Abraham's solvation parameters from the UFZ-LSER Database[80]. All of the physicochemical data used is available in the Data Repository.

To reflect differences between anthropogenically-driven shared socioeconomic pathways (SSP) and their influence on OPE fate in urban areas, we ran the model for an "SSP3-7.0" scenario using the back-calculated base-case emissions along with the projected difference in the temperature, windspeed and precipitation between the SSP1-2.6 and SSP3-7.0 scenarios in 2100. Data were obtained from the curated, quality-controlled CMIP6 projections available on the Copernicus Data Store[81]. We calculated ensemble-average decadal averages for 2041-2050 and 2091-2100 from all available model runs for each variable.

## Model application, sensitivity, and scenario analyses

We parameterized and applied the model in several different manners, depending on the intended purpose. First, we back-calculated the emissions from the measured air concentrations. For this, we parameterized the model using the averaged values of the leaf-area index, relative humidity, rain rate, windspeed, planetary boundary layer

height, and temperature across the ~3-month sampler deployment period at each location and annual-average values for 2018 for all other values. A key assumption of the model was that the air concentrations measured by the passive air samplers were representative of the urban areas across the sampling period. To test the applicability of this assumption, we ran the model using three different model boundaries (using the administrative boundary[82], and with a radius of 5 or 15 km from the sampling location) and compared the results for the emissions flux (kg m$^{-2}$) of each boundary. The modeled emissions for each of the boundary areas were within ± 2x of each other (Supplementary Table 2), well within our ± order-of-magnitude overall uncertainty, indicating that the fate processes within the city remained similar at different scales, and providing confidence that the model results could be extrapolated over a larger domain. Our estimates of total emissions used the cities' administrative boundaries under the assumption that those boundaries represented a cohesive unit across which emissions sources and fate were similar, while regressions with emissions proxies used the emissions flux (kg m$^{-2}$ yr$^{-1}$) from the 15 km buffer radius.

Second, to compare contaminant fate between cities, we ran the model using annual-average values for the sampler deployment year of 2018 with the estimated annual emissions described above to remove the influence of seasonality and show average differences between cities. We justify this because although air concentrations are known to vary in the course of a year[83,84], emissions of OPEs are thought to be driven more by the intensity of local sources than by seasonal effects, such as increases in vapor pressure at higher temperatures[83,85]. As discussed in Supplementary Results S2, we generally found that the factors indicated by our sensitivity analysis to control contaminant fate were poorly correlated with our estimated emissions, supporting the assumption that local sources controlled emissions was valid.

Third, to reflect differences between anthropogenically-driven shared socioeconomic pathways (SSP) and their influence on OPE fate in urban areas, we ran the model for an "SSP3-7.0" scenario using the back-calculated base-case emissions along with the projected difference in the temperature, windspeed and precipitation between the SSP1-2.6 and SSP3-7.0 scenarios in 2100. Data were obtained from the curated, quality-controlled CMIP6 projections available on the Copernicus Data Store[81]. We calculated ensemble-average decadal averages for 2041–2050 and 2091–2100 from all available model runs for each variable.

Fourth, we explored the influence of different parameters on the fate of chemicals across the "city-space" represented by different urban environments. For this, we defined two indices based on the area of urban film and of vegetation within a city. The first index defines how the built-environment impacts chemical deposition within a city. We defined this "sparsity index" (SI, m$^2$ m$^{-2}$) with Eq. (3):

$$SI = \log\left(\frac{A_{\text{cityfootprint}}}{A_{\text{film}} + A_{\text{vegetation}}}\right) \quad (3)$$

Where A$_j$ represents the area of compartment j in m$^2$. The second index explored the nature of those surfaces. We defined this "film-vegetation index" (FVI) as Eq. (4):

$$FVI = \log\left(\frac{A_{\text{film}}}{A_{\text{vegetation}}}\right) \quad (4)$$

For these scenarios we back-calculated emissions to a composite "average" city, consisting of the mean values for the city-specific variables not included in the SI and the FVI, targeting the mean concentration of each OPE across the 19 cities. We also conducted limited scenario analyses to test the response of the model to certain parameter sets. First, we tested the influence of the half-lives in the film and vegetation by multiplying the default value by 10 and 0.01,

respectively. Then, we tested the influence of climate by constructing a "low-deposition" scenario where the windspeed, temperature, and planetary boundary layer values were set to the 95th percentile across the 19 megacities, and the precipitation rate was set to the 5th percentile; and a "high-deposition" environment where the windspeed, temperature, and planetary boundary layer values were set to the 5th percentile across the 19 megacities, and the precipitation rate was set to the 95th percentile. For each scenario, we ran the model for 1000 SI values between −0.8 and 0.8, and for 1000 FVI values between −1.5 and 2.75, for a total of 1,000,000 model runs per scenario with the joblib v1.1 python package[86]. To generate the figures we plotted the magnitude of the largest fate process for mass leaving each city using a contour plot in matplotlib. The code to produce these plots is available in the archived version of our Data Repository.

Finally, we performed a model sensitivity analysis focused on the differences between cities to elucidate trends that control the fate of OPEs in different urban environments. We used the Elemental Effects[87] method to characterize MUM's sensitivity as it can identify non-monotonic, discontinuous interactions between variables. Although MUM is based on a system of linear equations, the parameters used to calculate the fugacity capacity can be non-linear and non-monotonic. We parameterized the range of values explored in the sensitivity analysis using the average location-specific values across cities and the observed inter-city ranges plus 10% on each side, a hypothetical chemical with average physical-chemical properties, and the observed ranges between chemicals plus 10% on each side, and the input probability distribution functions presented in ref. [8]. The Data Repository contains the parameterization for each input variable that was tested.

### Correlations with emissions proxies and controls

To investigate the sources driving OPE emissions, we correlated the log$_{10}$ transformed emissions flux using the 15 km$^2$ boundary area with various potential emissions proxies (such as GDP), and controls (such as temperature). Following initial correlations to reduce the number of variables, we correlated the emissions against the percentage of built-up area, bare area, and vegetated area for each city[60], the average temperature in each city across the sampler deployment period[68], the total population in each city[30], global GDP and GDP per capita[29,88], and against total and sector-specific CO$_2$ emissions from the Emission Database for Global Atmospheric Research (EDGAR)[31], which we used as proxies for emissions of OPEs from specific economic sectors. The extracted proxy and control values we used for each city are available in our Data Repository.

## Data availability

The data used and generated in this study have been deposited in the Borealis Data Repository with the DOI 10.5683/SP3/KT1DG5 (https://doi.org/10.5683/SP3/KT1DG5)[54].

## Code availability

The model and code used in this study are available either as a permanent version from this article's Data Repository (https://doi.org/10.5683/SP3/KT1DG5)[54] or from a repository on the lead author's Github (https://github.com/tfmrodge/FugModel), which also contains a tutorial for running the model.

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

## Acknowledgements

We would like to thank the Natural Sciences and Engineering Research Council (NSERC) of Canada for an NSERC postdoctoral fellowship to T.F.M.R., an NSERC Discovery Grant to A.G. (RGPIN-2018-04893), and an NSERC Discovery Grant to M.L.D. (RGPIN-2017-06654); and Environment and Climate Change Canada (ECCC) for Grants and Contributions funding to M.L.D., A.S., and T.F.M.R. in support of this project (GCXE22S058). We thank the United Nations Environment Program (UNEP) and the Chemicals Management Plan (CMP) for financially supporting the GAPS-Megacities study. We also thank the collaborators of the GAPS-Megacities study for their assistance in carrying out the sampling campaign.

## Author contributions

T.F.M.R.—Conception, investigation, data curation, methodology, writing (original draft), and visualization; A.G.—Conception, methodology, writing (review and editing), and supervision; M.L.D.—Conception, writing (review and editing), and supervision; E.G.—investigation and writing (review and editing); A.S.—Conception, writing (review and editing), and supervision.

## Competing interests

The authors declare no competing interests.
