## [Peer Review File · Nature Communications]

Emissions and Fate of Organophosphate Esters in Outdoor Urban EnvironmentsReviewer #1 (Remarks to the Author):

In this study, the Multimedia Urban Model was used to effectively evaluate the emissions and fate of OPEs in the global megacities' atmosphere within a good precision. The results improve our understanding of OPE migration and transformation in cities and urban design. The manuscript can be published after some revisions.

1.In the section, identifying drivers of OPE emissions, the author said, "driven by an r^2 of 0.31 for TCEP", but in Figure ED 2, the r^2 of TCEP seems to be 0.26 (TCIPP $r^2 = 0.31$). Is it a clerical error?

And I suggest the author check the consistency of data between Figure ED 2 and Table S3.

2.In the following discussion, concentrations for each of the OPEs were chosen to be the inputs. Meanwhile, OPEs are not in the POPs list. Therefore, it is suggested that the author modify the title of the article as "Fate of Organophosphate esters in Outdoor Urban Environments"?

3.Since A and B are marked in Figure 3, it is recommended to change the 'top' and 'bottom' in the figure title to A and B

4.In the section, model parameterization, the author predicted the impervious surface index using a linear regression ($r^2 = 0.78$). Would it be more rigorous to add p value?

Reviewer #2 (Remarks to the Author):

This manuscript aims to estimate the emissions and fate of OPEs in the atmospheric environment using the MUM and GAPS-Megacities network as a novel approach. The results are convincing and the structure of this manuscript is well organized. This manuscript may provide a further understanding of policy-making for careful urban design that can reduce the emission sources of PMT contaminants. Overall, this is a worthwhile study that deserves publication in Nat. Commun. I only have one comment: Line 324 - Are there data about the possible phytotransformation products of OPEs? If yes, it would be helpful to provide such information.

1 Where do they come from, where do they go? Emissions and fate of
2 OPEs in global megacities

3 Timothy F. M. Rodgers¹, Amanda Giang^{1*}, Miriam L. Diamond^{2,3}, Emma Gillies¹, Amandeep Saini^{4*}

4 *Reviewer comments are in italics*, our responses are in black normal font, **changed text is in blue**.

5 Reviewer #1

6 *In this study, the Multimedia Urban Model was used to effectively evaluate the emissions and fate of*
7 *OPEs in the global megacities' atmosphere within a good precision. The results improve our*
8 *understanding of OPE migration and transformation in cities and urban design. The manuscript can be*
9 *published after some revisions.*

10 *1. In the section, identifying drivers of OPE emissions, the author said, "driven by an r^2 of 0.31 for TCEP",*
11 *but in Figure ED 2, the r^2 of TCEP seems to be 0.26 (TCIPP $r^2 = 0.31$). Is it a clerical error?*
12 *And I suggest the author check the consistency of data between Figure ED 2 and Table S3.*

13

14 We thank Reviewer 1 for your kind remarks, and for your sharp eyes in pointing out our clerical
15 errors. Figure ED2 has been updated to match Table S3, which has also been corrected. Lines 125-128
16 now read:

17 **The strongest single correlation was with Σ GDP in the modeled area, which explained 36% of variation**
18 **(measured by r^2) for the $\log_{10} \Sigma_{10}$ OPEs, driven by strong correlations (r^2 of 0.31 – 0.19, $p < 0.05$) for, in**
19 **descending order, TmCP, TnBP, TCEP, TCIPP and TDCIPP (Figure ED 2).**

20 Further, the updated figure ED 2 and Table S3 are reproduced below:

21

22 Figure ED 1: Regressions between emissions proxies (Gross Domestic Product (GDP) at 2015
 23 purchasing power parity, anthropogenic emissions of greenhouse gases as CO_2 equivalents, and the
 24 proportion of anthropogenic CO_2 equivalent emissions from the Solvents and Products, Buildings
 25 and Non-Metallic Minerals (NMM) Production sectors, as described by EDGAR³³) and the \log_{10}
 26 transformed emissions flux ($\text{kg m}^{-2} \text{yr}^{-1}$). Selected regressions with $p < 0.05$ are shown, r^2 is the
 27 adjusted correlation coefficient. Table S3 shows all regressions with $p < 0.05$.

28

29 Table S1: Regressions with $p < 0.05$ between emissions proxies (Σ GDP, Gross Domestic Product (GDP)
30 at 2015 purchasing power parity, GDP per capita at 2015 purchasing power parity, Σ CO₂ Emissions,
31 anthropogenic emissions of greenhouse gases as CO₂ equivalents, and the proportion of
32 anthropogenic CO₂ equivalent emissions from the Solvents and Products, Energy for Buildings, Solid
33 Waste Incineration, Non-Metallic Minerals (NMM) Production, and Combustion for Manufacturing
34 sectors, as described by EDGAR³¹), and transport controls (relative humidity %, precipitation rate
35 mm/hr, area of vegetation m²) vs the log₁₀ transformed emissions flux (kg m⁻² yr⁻¹) for the specified
36 OPE.

Regression	Independent Variable	m1	b	p_m1	p_b	r ²	adjr ²	p_reg
Σ OPEs	Σ GDP	2.46E-12	-5.98	3.72E-03	1.94E-17	0.40	0.36	3.72E-03
TCIPP	%CO2 Solvents and Products	3.06E+01	-6.29	4.69E-03	7.00E-18	0.38	0.35	4.69E-03
TBOEP	RH	2.97E-02	-9.01	0.01	2.44E-09	0.36	0.32	0.01
Σ OPEs	GDP per Capita	2.00E-05	-5.91	0.01	1.73E-17	0.35	0.31	0.01
TmCP	Σ GDP	2.84E-12	-8.04	0.01	8.80E-18	0.35	0.31	0.01
TnBP	Σ GDP	1.68E-12	-7.80	0.01	2.61E-21	0.34	0.30	0.01
TCIPP	%CO2 Energy for Buildings	2.89E+00	-6.50	0.01	2.23E-15	0.32	0.28	0.01
TCEP	GDP per Capita	1.90E-05	-6.42	0.02	1.32E-17	0.30	0.26	0.02
TCEP	Σ GDP	2.19E-12	-6.45	0.02	3.31E-17	0.30	0.26	0.02
Σ OPEs	RH	2.08E-02	-6.97	0.02	5.16E-10	0.30	0.25	0.02
TEHP	%CO2 Solid Waste Incineration	-1.41E+02	-8.40	0.03	3.57E-17	0.29	0.24	0.03
TmCP	GDP per Capita	2.22E-05	-7.95	0.02	9.05E-18	0.28	0.24	0.02
TCIPP	RH	2.35E-02	-7.54	0.02	1.97E-09	0.28	0.23	0.02
TBOEP	GDP per Capita	2.35E-05	-7.46	0.03	9.88E-15	0.28	0.23	0.03
TCIPP	Σ GDP	2.37E-12	-6.34	0.02	4.57E-16	0.27	0.23	0.02
TmCP	Precipitation Rate	3.72E+00	-7.96	0.02	1.91E-17	0.27	0.23	0.02
Σ OPEs	%CO2 Energy for Buildings	2.21E+00	-5.97	0.03	1.57E-15	0.26	0.21	0.03
TDCIPP	Σ CO2 Emissions	9.46E+04	-7.81	0.03	2.35E-16	0.25	0.21	0.03
TPrP	Area of Vegetation	-1.10E-10	-7.41	0.03	3.42E-22	0.25	0.21	0.03
TBOEP	Precipitation Rate	4.04E+00	-7.44	0.04	1.69E-14	0.25	0.20	0.04
TCEP	Σ CO2 Emissions	7.49E+04	-6.38	0.03	2.94E-17	0.24	0.19	0.03
TDCIPP	Σ GDP	2.40E-12	-7.84	0.04	6.71E-16	0.24	0.19	0.04
TnBP	Σ CO2 Emissions	5.29E+04	-7.72	0.04	4.30E-21	0.23	0.19	0.04
TCIPP	GDP per Capita	1.88E-05	-6.27	0.04	3.48E-16	0.23	0.18	0.04
Σ OPEs	%CO2 Solvents and Products	2.02E+01	-5.78	0.04	1.53E-17	0.23	0.18	0.04
Σ OPEs	Σ CO2 Emissions	7.01E+04	-5.83	0.04	9.76E-17	0.22	0.18	0.04
TCIPP	%CO2 NMM Production	-6.28E+00	-5.53	0.04	2.74E-15	0.22	0.17	0.04
TmCP	RH	2.21E-02	-9.05	0.04	5.21E-10	0.22	0.17	0.04
TmCP	Σ CO2 Emissions	8.56E+04	-7.89	0.04	2.14E-17	0.22	0.17	0.04
TEP	%CO2 Combustion for Manufacturing	-2.22E+00	-7.23	0.05	1.98E-16	0.22	0.17	0.05
TPhP	Σ GDP	1.95E-12	-7.38	0.05	2.44E-17	0.21	0.16	0.05

37

38

39 *2. In the following discussion, concentrations for each of the OPEs were chosen to be the inputs.*
40 *Meanwhile, OPEs are not in the POPs list. Therefore, it is suggested that the author modify the title of the*
41 *article as “Fate of Organophosphate esters in Outdoor Urban Environments”?*

42 We have changed the title as suggested, line 135 now reads:

43 Fate of Organophosphate Esters in Outdoor Urban Environments

44 *3. Since A and B are marked in Figure 3, it is recommended to change the ‘top’ and ‘bottom’ in the figure*
45 *title to A and B*

46 We have made the change as suggested, the figure caption now reads:

47 Figure 3. A: “City-space” figure showing the dominant chemical fate process for the \$\sum_{10}\$ OPEs in a
48 hypothetical “average” city with their built-environments described by the surfaces vs film-vegetation
49 indices (as described in the main text). Contour colors show how the dominant fate process for this
50 “average” city varies across these two indices, with the intensity the proportion of total emissions
51 undergoing that process (as labelled). Points show where the 19 GAPS-Megacities locations fit on these
52 axes; the color of each point represents the dominant chemical fate process in each city using its 2018
53 parameterization.

54 B: \$\sum_{10}\$ OPE fate diagrams for the “sparse,” “densely vegetated,” and “densely urbanized” archetypical
55 cities of Cairo, Bogotá, and Kolkata, respectively, for 2018. Dashed lines represent transformation
56 processes, solid lines transport processes. Emissions (\$\text{kg yr}^{-1}\$ ) are shown entering the lower-air
57 compartment and fate process values are given as the % of total emissions. Values shown on each figure
58 may not sum to 100 as only larger processes are shown, see Figure ED 3 for fate diagrams with all
59 processes.

60 *4. In the section, model parameterization, the author predicted the impervious surface index using a*
61 *linear regression ($r^2 = 0.78$). Would it be more rigorous to add p value?*

62 We have added the significance of the regression to line 646 and SI Figure S1 as follows.

63 ...ISI using a linear regression (\$r^2 = 0.78\$, \$p < 0.01\$ ) with the “built-up area density”

64

65 Figure S1: Relationship between the impervious surfaces index and population density in the built-up
 66 area (as described in the Methods). The brackets indicate the model boundary parameterization
 67 represented in each point, a lack of brackets indicates the city administrative area.

Reviewer #2

Comments:

This manuscript aims to estimate the emissions and fate of OPEs in the atmospheric environment using the MUM and GAPS-Megacities network as a novel approach. The results are convincing and the structure of this manuscript is well organized. This manuscript may provide a further understanding of policy-making for careful urban design that can reduce the emission sources of PMT contaminants.

Overall, this is a worthwhile study that deserves publication in Nat. Commun.

I only have one comment: Line 324 - Are there data about the possible phytotransformation products of OPEs? If yes, it would be helpful to provide such information.

Thank you for your kind remarks. We have expanded on the discussion of plant phytotransformation on Lines 210-223 as follows, with one new reference listed below:

For OPEs and other compounds with shorter transformation half-lives in vegetation (i.e. that were susceptible to phyto-transformation), plants acted as fixing and transforming surfaces, reducing the concentration of OPE parent compounds that either remained in the air compartment or were exported to aquatic ecosystems. Although direct atmospheric transformation products of OPEs can be more persistent and toxic than the parent compounds,³⁹ plants have been shown to rapidly transform the predominantly triester OPE parent compounds primarily through direct dealkylation to diester products, or through hydroxylation to hydroxylated OPEs.⁴⁰ Subsequent transformation of the diester products has been observed for the non-chlorinated OPEs.³ This continued metabolization suggests that plant transformation may be able to reduce the overall persistence of non-chlorinated OPEs and their transformation products, thereby lowering the overall hazard posed from OPEs deposited to plants. By contrast, for the chlorinated OPEs the lack of continued metabolization indicates that the transformation products may continue to be problematic. For two cities (Bogotá and Mexico City), reaction in the soil dominated overall fate of TPhP, following chemical deposition to vegetation and subsequent wash-off to soil, as TPhP is less susceptible to transformation in vegetation than in soil.

39) Zhang, Q.; Yao, Y.; Wang, Y.; Zhang, Q.; Cheng, Z.; Li, Y.; Yang, X.; Wang, L.; Sun, H. Plant Accumulation and Transformation of Brominated and Organophosphate Flame Retardants: A Review. *Environmental Pollution* **2021**, *288*, 117742. <https://doi.org/10.1016/j.envpol.2021.117742>.